# Support over Social Media among Socially Isolated Sexual and Gender Minority Youth in Rural U.S. during the COVID-19 Pandemic: Opportunities for Intervention Research

**DOI:** 10.3390/ijerph192315611

**Published:** 2022-11-24

**Authors:** Sana Karim, Sophia Choukas-Bradley, Ana Radovic, Savannah R. Roberts, Anne J. Maheux, César G. Escobar-Viera

**Affiliations:** 1Learning Sciences Research, Digital Promise 1001 Connecticut Ave NW #935, Washington, DC 20036, USA; 2Department of Psychology, Dietrich School of Arts & Sciences, University of Pittsburgh, Pittsburgh, PA 15260, USA; 3Division of Adolescent and Young Adult Medicine, University of Pittsburgh School of Medicine, UPMC Children’s Hospital of Pittsburgh, Pittsburgh, PA 15260, USA; 4Center for Enhancing Treatment & Utilization for Depression & Emergent Suicidality (ETUDES), University of Pittsburgh School of Medicine, Pittsburgh, PA 15260, USA; 5Department of Psychiatry, University of Pittsburgh School of Medicine, Pittsburgh, PA 15260, USA

**Keywords:** rural youth, social isolation, LGBT youth, social media interventions, social media

## Abstract

Sexual and gender minority (SGM) rural adolescents are at risk for higher levels of social isolation, a well-known risk factor for depression and other negative health outcomes. We qualitatively examined how rural SGM youth seek emotional and informational support, which are protective factors for social isolation on social media (SM) regarding their SGM identity, and determined which SM platforms and tools are most effective in providing support. We conducted semistructured online interviews with rural SGM teens who screened positive for social isolation in spring 2020 and used a thematic analysis approach to analyze the data. Sixteen youths participated in interviews. Themes included seeking emotional support through SM groups and communities, seeking emotional support in designated online SGM spaces, using SM feeds for informational support, and disclosing SGM identity differentially across platforms. SM-based interventions could be leveraged to provide emotional and informational support for rural SGM youth across specific SM platforms and consider whether they are providing emotional or information support. Interventions focused on informational support may best be used on content-based platforms. Those designed to combat social isolation and connect marginalized SGM youths to similar others might benefit from community and forum-based platforms.

## 1. Introduction

Social isolation can be defined as an absence of social interactions, contacts, and relationships with other people at multiple levels of human interaction, including individual, group, and community [1], and has been linked to poorer mental health outcomes [2,3,4]. Geographic isolation, rurality, and identifying as sexual and gender minority (SGM; e.g., lesbian, gay, bisexual, transgender, non-binary, queer) increase the risk for social isolation [5]. These experiences have increased globally across different age groups during and after the COVID-19 related lockdowns [6,7,8,9,10], raising concerns about their impact on mental health and calls to action about intervening on social isolation as experienced by youth [11]. SGM youth are those whose sexual orientation includes attraction towards same-gender people and/or whose gender identity is different than their assigned sex at birth [12], and are at higher risk of social isolation than their non-SGM peers, especially SGM youth living in rural areas. While community-related resilience, support, and connectedness protect against social isolation [13,14,15,16,17], in these close-knit communities, familiarity and sameness are valued and obtaining support, resources, and finding connectedness can be difficult for SGM youth, whose identities do not conform with the prevailing cultural and social norms in rural communities [18,19]. This, combined with the geographic isolation of many rural communities and COVID-19 lockdowns, created a compounded situation of isolation for rural SGM youth, impacting their ability to seek and find support within an already difficult physical environment.

As of 2022, 97% of teenagers use the Internet daily [20], using web-based or mobile tools that foster communication [21] to become part of a community, find relevant information [22], and seek emotional and informational support that users may not be able to access face-to-face in their geographic location [22,23]. These types of support are particularly important to SGM youth, who may also lack community mental health resources due to their geographic isolation, and may put these youth in need for safe spaces and connection to youth with similar identities [24]. The lack of accessible SGM-specific health resources and information in their communities may make rural SGM youth more vulnerable to negative mental health outcomes than their non-SGM peers in the same areas, and compared to both SGM and non-SGM youth in urban settings [25]. Importantly, empirical evidence shows that positive representation of and establishing connections with other SGM youth that share the same intersectionality (i.e., SGM and rurality) are important sources of perceived support for SGM youth living in rural areas [26]. 

While minority stress theory explains the existing disparities in negative mental health outcomes among SGM people via overt and internalized discrimination, it also posits that connecting with like-minded communities may act as a protective factor against these negative outcomes [27]. Social media is one way to provide this necessary outlet due to the high proportion of SGM youth that use social media platforms [22]. Social media-delivered interventions developed as a means of emotional and informational support, mental health education, and resources for young people might help those (such as rural-living SGM youth) who experience social isolation and associated mental health problems [28,29] and may not have the opportunity to explore their identity [25], connect with others like them, receive care, and gain access to SGM resources in their offline lives [23]. However, rural SGM youth have expressed experiences of finding both supportive communities and online harassment online [30] and there is a paucity of research on social-media-delivered mental health interventions specifically addressing the needs for rural-living SGM youth. 

We sought to qualitatively explore (1) how rural-living SGM youth used existing social media platforms for seeking support during the first COVID-19 lockdown, as well as which platform characteristics and features contributed the most to accomplish this purpose and (2) how digital interventions could be delivered on specific social media platforms to provide support to rural-living SGM youth. While we did not report in this paper about the needs of social support and life situations of this group of rural SGM youth, these were reported elsewhere [27]. 

## 2. Materials and Methods

### 2.1. Sample Selection

All recruitment and data collection procedures were approved by the University of Pittsburgh Institutional Review Board. We obtained a waiver of parental permission to protect the privacy of youth who may not have otherwise been ready or willing to be out to their parent or caregiver. We recruited SGM adolescents living in rural areas of the United States using purposeful sampling via advertisements on Instagram and Facebook between February and April 2020. Social media facilitates recruitment of groups commonly underrepresented in research [31,32], and purposeful sampling involves identifying and selecting participants that personally experience the phenomenon of interest [33]. We utilized an advertisement creation feature on Instagram and Facebook to limit the advertisement reach to zip codes classified by the Federal Office of Rural Health Policy in the Health Resources Service Administration as rural areas [34]. Recruitment also involved targeted ads based on topics of interest for users on social media (e.g., LGBTQ rights). 

Eligible youth were 14–19 years of age; screened for perceived social isolation; lived in a rural zip code; identified as gay, lesbian, bisexual, transgender, or queer; owned and used at least one profile on social media; and spoke English. We used a 4-item measure developed by the Patient-Reported Outcomes Measurement Information System (PROMIS) of the National Institutes of Health initiative [35] and followed PROMIS’s recommendation to consider those scoring 16 and higher as positive. Potential participants received an email with a link to online informed assent/consent forms and instructions to download and install a video conference application for the online interviews that complies with the Health Insurance Portability and Accountability Act of 1996 (HIPAA) to protect sensitive patient health information [36]. Participants were compensated with online gift cards for their time. 

### 2.2. Data Collection and Structured Interviews

A social science researcher with psychiatry training who identifies as a cisgender man conducted all online interviews. Interviews were 60–75 minutes long. We developed a structured interview guide (see Appendix A) based on both previous knowledge and our own research on this topic. There were three main topics covered: (1) participants’ top three social media platforms and their behaviors when using these (“Tell me more about the activities you usually engage in on”); (2) LGBTQ-related topics they would like to learn more about (“Describe to me what topics or subjects related to growing up as an LGBTQ person in rural area you want to learn more about. These could include stuff like the process of coming out or how to use your social media in a healthier way, for example.”); and (3) preferred ways to receive that information (“Tell me how you would like to receive information about these topics”). Interviews were audio-recorded and transcribed verbatim. Transcripts were entered into NVivo 12 [37] for analysis. 

### 2.3. Analysis

We used a reflexive thematic analysis approach with an experiential framework [38], in which we focused on describing the lived experience of youth participating in the study with a constant check on our assumptions and practices that might influence our interpretation of the data. We followed guidance from Nowell et al. [39] to improve rigor, where the participant was the central unit of analysis. This was appropriate because the identified themes were driven by SGM youth’s explicit meaning and experiences [40]. We first created primary codes for each social media platform mentioned by participants and coded all instances where participants mentioned that specific platform. Next, we used a hybrid deductive and inductive approach and further developed codes based on the existing codebook. This method was specifically applied to those codes falling under the social media platform category where mentions of specific social media platforms were referenced. We chose this approach due to its rigor [41] and because it allowed us to leverage previous research of social media use that may prevent negative mental health experiences [42]. We used inductive coding for topics related to privacy, outness, and community-seeking on social media platforms due to adolescents mentioning these topics on their own accord, spontaneously, and frequently. To increase trustworthiness of our analytic process and to generate a final codebook, we circulated and triangulated our codebook among the authors, who have qualitative expertise and come from interdisciplinary fields of public health, medicine, and psychology. After completing this process for four transcripts, we agreed upon the final codebook, continued coding the remaining transcripts, and identified and named themes during team meetings. 

## 3. Results

### 3.1. Participant Characteristics

Sixteen rural-living SGM adolescents who screened positive for social isolation completed the interviews. The average age was 17.3 years. Full demographics are available in Table 1. A breakdown of the social media platforms that participants mentioned that they used and their characteristics can be viewed in Table 2. Participants mentioned 12 social media platforms in total. When listing their top three platforms, 10 participants mentioned YouTube (62.5%), 9 said Facebook (56.3%), 6 said Instagram (37.5%), and 7 said Reddit (43.8%). Pseudonyms were used for all participants to ensure anonymity. 

As one participant, Jackson (19, bisexual, transgender male), can summarize about social media use and support: “I’d say different social media platforms can give different degrees of support…it just depends on where you are, which social media platform to the level of support you would find”. We identified twelve unique codes related to rural SGM youth, social media use, and support (see Appendix B for full codebook). We report four main themes and relevant quotes below.

### 3.2. Outness Depending on Platform

Outness, or how open participants were about their SGM identity, varied depending on the social media platform and influenced youth’s ability to be “out” and fully themselves. Social media platforms where some participants were completely or majorly out included Reddit, Discord, and Tumblr. Tony, a 17-year-old lesbian transgender young woman, had mentioned that they had met their current foster mother through Tumblr: “Well for one, I met my foster mom on social media, on Tumblr, which is fantastic, and I couldn’t be happier.” Cyrus, a 17-year-old gay cisgender young man, describes how using Discord groups (also colloquially known as “servers”) gave him the opportunity to meet others who identify as SGM, and in turn, shaped his sense of self:


*When I first started going on social media and stuff like that, I joined on a lot of Discord groups and stuff, like LGBTQ people and it’s really just made me into the person I am today.*


Participants mentioned they were not as likely to be completely out on social media platforms where they connected with a large number of people they already knew in their “offline” life. For example, those who spoke about Facebook had both offline people (persons participants interacted with outside of social media) and family members as Facebook friends. Participants also had varying experiences with how “out” they were to those that they had added on Facebook.

Similar to Facebook, participants who talked about Instagram described being out with varying degrees. Family members were a frequently mentioned group that participants said prevented them from being fully out. However, this variability depended on what offline people participants were connected with on social media platforms. As Alexi, an 18-year-old lesbian cisgender young woman, describes her experience on other social media platforms versus Instagram:


*There are certain family members I’m still not out to at this point in time, so I’m more quiet about that on certain platforms. There are some platforms I don’t have family members on and I will share maybe informative posts, particularly on Instagram. That’s the platform I use most for that. Other than that, not really.*


### 3.3. Joining Groups and Communities to Seek Emotional Support

Participants overwhelmingly mentioned using social media to seek emotional support regardless of the platform. Specifically, participants reported frequently seeking support in online communities and groups, where users can create, interact, and post content in forums developed based on similar interests. Grant, a 16-year-old gay cisgender young man, mentioned how he utilized the group chat feature on Snapchat: 


*Snapchat groups. Those people I know and I’m friends with. Even though we’re spread out, we’ll go to the group and we’ll message, ask how their day’s been.*


Another participant, Allie, a 15-year-old bisexual cisgender young woman, discussed the benefits of Facebook groups:


*I can feel welcome so I feel belonged and so I can learn. And generally when I join a group, it’s because it’s something that I’m interested in, something that I want to learn.*


Participants who spoke about seeking specific emotional support would look for those to talk to when they were not feeling well emotionally or were having problems. For example, there was one social media platform, Vent, that one participant named Tony used to seek emotional support by “venting” as the platform was designed to be used, although no other participant mentioned the platform. Elliot, a 19-year-old bisexual and transgender young person, spoke about losing an online community group and its negative impact on them, as it gave them access to a community from parents who restricted their Internet use:


*If you didn’t have accepting parents, you’re just kind of screwed. My parents blocked a lot of the internet for most of my growing up. Tumblr got through the cracks, and I learned a lot then. Then they found out about it and blocked it. I lost a big community thing.*


Another example of community and group-seeking included Facebook, where participants would use Facebook Messenger—Facebook’s integrated private messaging system between approved friends mutually “added” on the platform—and Facebook groups, where users can expand their interactions on Facebook to those they do not have “added”, to seek support. Unlike Facebook, Reddit allows for more anonymity through the use of usernames and no required profile photo. Those who used Reddit reported seeking out subreddits (forums designed for users with similar interests and identities to interact) to receive advice.

In addition to participants using groups to seek emotional support, participants would also seek out groups created for SGM youth. However, youth seemed to do this with the intention of connecting and seeking emotional support from others who identified as SGM youth. For example, three participants used the “LGBTeens” subreddit on Reddit. Cyrus mentioned looking for servers on Discord, a platform similar to Reddit in its anonymity features and individual channels depending on interest, where there were others who identified as SGM. Dustin, an 18-year-old gay cisgender young man, sought out the SGM community on the Amino platform, another anonymous, username-based platform allowing users to join communities with similar interests to theirs. He mentions:


*There would be people that would be like, ‘Oh, thank you so much. This post really helped me’. Like, ‘I’m definitely going to use this, this brightened up my day.’ And I would have people message me and I don’t know, I thought those were positive experiences.*


Participants would utilize groups and also “friend” (add) people that they do not know offline on Facebook, especially others who also identify as SGM. Allie describes her experiences with joining SGM groups on Facebook as a way to connect with others who identified as SGM.


*Well at first, I was new to this whole community thing and I wasn’t sure of where I would be and if I would be welcomed, you know? So I joined a couple groups on Facebook and such like that to connect with other people who are in the community and kind of get to know what it’s all about.*


### 3.4. Using Social Media Platform Feeds to Seek SGM-Specific Informational Support

Participants also discussed how they used social media platforms to seek information and educate themselves about LGBTQ issues, news, and representation. Twitter was the only platform mentioned where participants sought more general information such as news and politics on their timeline from accounts they followed, although participants did not mention what news and political content they consumed. Although not actively seeking information about SGM news, Dustin spoke about how he found it interesting to hear about SGM rights and enjoyed hearing other people talk about LGBTQ topics instead of engaging with it. Lily, a 19-year-old bisexual and transgender young person, used Tumblr for SGM activism and research.

Participants also mentioned Reddit threads for information seeking, including looking for information that may be associated with their SGM identity such as learning about top surgeons (those performing procedures to alter the amount of breast tissue on one’s body) and mental health and medical resources. Two participations mentioned that they used Tumblr to learn more about SGM research and to find others with experiences like them. Payton, a 19-year-old lesbian cisgender young woman, also spoke about using TikTok’s “For You Page” to find information:


*Well, on Twitter and TikTok, I follow a lot of LGBTQ members and because of that, it seems like the app targets that interest and I constantly see other new LGBTQ members and I’m pretty open about it on there and I just follow the content.*


Several participants mentioned seeking out SGM content created by others who identified as SGM sharing their own experiences. Platforms where participants would look for SGM content included TikTok, Tumblr, Twitter, and YouTube. This type of content included seeking others who identified as SGM, such as watching YouTube influencers and coming out videos, where the uploader addressed having an SGM identity to a large audience for the first time. Jackson, a 19-year-old bisexual transgender female, mentioned watching YouTube to see “other trans people in how they’re doing in their struggles.”

Some participants would primarily seek out SGM informational support more in a passive manner (defined as consuming social media without directly interacting with others, such as watching videos) and by scrolling on their “home feeds”, the default page on most social media platforms that includes content from people whom a user follows and recommended content based on algorithms. For example, these participants did not appear to post questions on news feeds asking for information. As Lily describes:


*Well, I prefer to find those pieces of information while scrolling through my newsfeed, similar to Facebook or Tumblr. I prefer to read them over text or to find a short video.*


### 3.5. Utilizing Privacy Features for Wellbeing and Support

Privacy is defined here as the amount of personal information that participants made publicly accessible. This includes whether their profile photo is of themselves, whether their first or full name is visible on their account and/or in their username, whether their account is public or “locked”, and how they determine whose friend request to accept.

Participants took advantage of the privacy features on their preferred platforms that allowed them to be selective in who could see their content. For example, participants mentioned using the “Close Friends” aspect of Instagram. Although any of a user’s followers (or anyone in general if the user has a public account) may view their “stories” (content uploaded on their account that is viewable for up to 24 h), the “close friend” feature allows the user to restrict access to parts of their “stories” to only certain followers. Participants mentioned using this feature for seeking support from a group of peers of their choosing. As Winter, a 14-year-old bisexual transgender young person, describes: 


*On Instagram, you can have the option to choose Close Friends, and you can post just things for only your Close Friends to see. So I put up things that are basically like, “I need somebody to talk to right now.” And I get a lot of positive people saying like, “Hey, how can I help?*


As similarly discussed in previous sections, participants actively used group chat features where they could be more open about themselves because of the limited viewing of content within the group. For example, instead of mentioning the photo-sharing feature and public “stories” that Snapchat is more known for, participants discussed using group chats and sending text messages through the chat feature.

Three participants mentioned having more than one account on a single social media platform as a way of curating multiple social media environments, in order to have anonymity on the same platform with a different use experience with more privacy. Two participants described these alternative accounts as important for their wellbeing. Emily, a 19-year-old bisexual cisgender female, had another account on Facebook with all of her personal information removed: “they’re just a place that I could connect with people that I knew that my family wouldn’t find”. Tony, a 17-year-old lesbian and transgender young person, had two additional blogs on Tumblr outside of their “main one”, one for positivity, and the other as a mental health blog to open up and “vent” about issues they were dealing with.

For platforms such as Reddit and YouTube, participants were not likely to have any identifiable information about themselves at all, nor did they have others added as “friends” on them. Tony describes their experience:


*It’s pretty impersonal because even if I respond to someone else’s comments, I didn’t take the time to look at their username or click their profile or anything. It’s just about the specific content that they posted in the community that I’m in. I’d say that’s how it works for pretty much a majority of Reddit.*


## 4. Discussion

### 4.1. Initial Findings

We interviewed rural-living SGM youth who felt socially isolated to explore how they used social media platforms to seek support during the first COVID-19 lockdown. Our results provide an understanding of how rural SGM youth use different social media platforms to seek informational and emotional support, and how these platforms could be leveraged to provide SGM-specific support. Further, we provide recommendations that might be helpful for researchers seeking to develop interventions for providing support to rural-living SGM youth over social media. 

### 4.2. Passive Seeking of Informational Support

Our findings confirm and expand previous research in that consuming and receiving informational support in a passive manner plays a role in adolescent identity development, as it allows adolescents to feel seen and represented by viewing others similar to them [50], particularly for those identifying as SGM youth [51]. Participants did not appear to actively ask others for informational support over social media. Most consumption of information and news, including matters regarding SGM issues, was performed passively over the use of video and posts. 

YouTube was the most popular platform for seeking informational support, and combined with the current popularity of TikTok, video-based content may be the best method to deliver information. SGM adolescents have expressed in previous studies how video content on social media can be considered an educational tool [24]. Moreover, the visibility of SGM creators and the way information is disseminated via video is easily accessible. However, depending on the content being delivered, popular platforms that predominantly involve passive use such as YouTube and TikTok may have negative effects, with the potential to trigger or worsen depressive symptoms [50]. Researchers working on social media-delivered interventions for health and wellbeing may want to intervene to suggest content on the platform that are specific and culturally tailored to rural SGM youth. Importantly, given that increased social isolation among youth is a phenomenon reported globally [11], intervention content must incorporate not only content specific to rural-living SGM youth but also tailor this content to different national and cultural contexts.

### 4.3. Active Seeking of Emotional Support

Our results align with and expand the tenets of minority stress theory, in that connecting with like-minded communities (on social media) is a source of perceived emotional support, and therefore, it might serve as protective factor for mental health among marginalized youth [28]. SGM youth socialized with peers on social media as a means to obtain emotional support. Rural SGM youth appeared to seek two forms of emotional support: general support for wellbeing and SGM-specific support in designated SGM youth social media groups. Some platforms were used to meet and seek support from people that adolescents do not know in their offline life, while other platforms were used to keep in touch or maintain relationships with established support systems. While previous research has found that emotional support can help reduce social isolation and loneliness for adolescents in general, those with marginalized identities may reap the greatest benefits from connecting and receiving emotional support from others with similar identities [52,53]. Research has also found that SGM youth use social media to join close-knit communities to reduce social isolation more often than their heterosexual counterparts [54]. Similarly, rural SGM youth participants in this study spoke about seeking out and adding others who also identified as SGM, suggesting an interest in connecting with others who share their SGM identities. 

Using this information, future work on social media-delivered interventions could prioritize connecting rural SGM youth to others similar to them. Rural SGM youth appear to prefer social support compared to other forms of support given their stage of development as adolescents [55]. The use of social spaces such as forums and groups designated for rural SGM youth similar to Reddit subreddits, Discord servers, and Facebook groups may be beneficial due to their community aspect and ability to balance privacy and social connection, and can include prompting questions for participating youths to answer about the emotional support that they may want to receive. Group chats were also frequently mentioned if they were able to be utilized through the platform; interventions can utilize group chat features to connect rural SGM youth so they can share their experiences with each other and give and receive emotional support. 

### 4.4. Preferred Social Media Platforms among Rural-Living SGM Youth 

Participants used SGM-specific groups on existing platforms than platforms specifically designed for SGM youth. However, these existing platforms were those where their personal information was not as publicly accessible, such as secondary accounts with restrictive privacy features on Instagram, nor were they likely to have offline people added, such as Reddit and Discord. Rural SGM youth may prefer these existing platforms because they already have accounts created and do not need to create new ones or because established platforms offer access to a larger group of potential peers. Research has shown that social media platforms that offer anonymity and the ability to deidentify oneself are preferred by those experiencing mental illness symptoms, providing a space to discuss mental health [56]. 

Adolescents also understood how to use filter and block options on social media platforms to choose who could see their content. These features were used to manage visibility of personal information, including sexuality and gender identity, and their discussion of negative experiences for which they were seeking support. This includes the “Close Friends” feature of Instagram, where users can create a list of followers who are allowed to view their private Instagram stories, and Facebook lists. Most social media platforms, including Facebook, Instagram, Twitter, and TikTok, allow users to make their accounts private to the public and block specific users, even if those users are following them. The use of privacy features may act as a form of safety measure and allow for the opportunity for selective visibility for SGM youth [57]. However, researchers working on developing interventions must also be aware of the lack of regulation and privacy violations that larger social media companies and platforms are frequently criticized for [58,59] to ensure the full safety and trust of rural SGM youth.

Rather than developing new platforms, we recommend that those developing interventions for rural SGM youth use existing platforms. For safety, confidentiality, and trust, these interventions should take advantage of privacy features offered by said platforms. Platforms such as Facebook allow groups to be made private, locked, and requiring permission to join. Similarly, Discord servers can also be made private and require requested permission to participate. Other platforms such as Instagram also feature a “vanish mode” in their private messaging system that allows all messages sent in the private chat to disappear, which interventions can use to remind rural SGM youth to use to erase their conversation for their safety. 

### 4.5. Limitations

Our work has several limitations to consider. For example, the majority of participants identified as White, and a large number of participants lived in the Midwest and South compared to other regions in the United States. The experiences of racially marginalized SGM youth, particularly in rural areas, can differ tremendously, and it is important that future studies prioritize their voices. Although rural spaces in the United States have larger White populations, this often results in excluding and forgetting racially marginalized groups in these areas. We also did not ask participants to give their preferred pronouns, which may have affected how they were represented in this analysis. More broadly, this study specifically focused on rural SGM populations in the United States, whose experiences cannot be assumed to be the same as rural SGM populations in other countries and contexts. Finally, the small sample size does not allow us to make substantial conclusions about how the majority of rural SGM youth use social media.

Social media habits and use are constantly changing; as such, the timing of the interviews also serves as a limitation. Interviews with rural SGM youth were completed in late spring of 2020, when COVID-19’s impact in the United States was still beginning. This period of time when nearly everything was performed virtually, including socializing via social media, may have affected participant responses and views. Additionally, this may explain the seldom mentions of TikTok, with only two participants mentioning the platform and one participant talking about it at length. The platform did not gain tremendous popularity until later in the year; future studies are needed to examine how a larger sample of American rural SGM youth use TikTok. Despite these limitations, we have been able to qualitatively analyze how an underrepresented population of rural SGM youth in the United States may benefit from targeted interventions over social media to reduce social isolation.

## 5. Conclusions

Social media-delivered interventions designed to prevent social isolation among rural SGM youth should consider how a platform serves SGM youth’s needs for a sense of community, social and informational support, and connections with similar others. Future research on these interventions should consider what degree of socialization and education researchers and SGM youth-oriented programs are trying to convey with their programs for rural SGM youth. The number of youths identifying as SGM and their needs have been gaining visibility, and it is important to take advantage of social media and technology in order to provide the safety, support, and knowledge that youth may not otherwise have access to, especially those in marginalized and isolated communities. Using the information provided by rural SGM youth regarding their use of social media platforms, we believe that mental health interventions targeting rural SGM youth on social media for this population could benefit from distributing interventions over social media as well. Intervention developers should also be mindful about the platform chosen and promote further incentives for larger studies to look at how rural SGM youth use social media. 

## Figures and Tables

**Table 1 ijerph-19-15611-t001:** Self-Identified Demographics (*n* = 16).

Characteristics	*n*
Gender Identity	
Transgender	6
Cisgender	10
Sexual orientation	
Bisexual	7
Gay or lesbian	9
Race (*n*)	
White, non-Hispanic	14
Hispanic	1
Pacific Islander	1
Region (*n*)	
Midwest	6
Northeast	3
South	6
West	1
Age (mean years)	17.6

**Table 2 ijerph-19-15611-t002:** Social Media Platform Characteristics.

Platform	Description	Year Released
Amino	Platform featuring standalone, user-generated communities to discuss specific interests. Users can scroll through curated posts, message privately or in public chat rooms and see posts from their following list [43].	2014
Archive of Our Own (AO3)	Noncommercial and nonprofit central hosting site for transformative fanworks such as fanfiction and, in the future, other transformative works such as fanart, fan videos, and podfic [44].	2007
Discord	Chat platform with user-generated topics (servers) [45].	2015
Facebook	Connects people with friends, family, acquaintances, and businesses from all over the world and enables them to post, share, and engage with a variety of content such as photos and status updates [45].	2005
Instagram	Photo sharing application that lets users take photos, apply filters to their images, and share the photos instantly on the Instagram network and other social networks [45].	2010
Pinterest	photo sharing social network that provides users with a platform for uploading, saving, and categorizing “pins” through collections called “boards”. Boards are typically organized by theme [45].	2010
Reddit	Social news site that contains specific, topic-oriented communities of users who share and comment on stories [45].	2005
Snapchat	Allows users to send and receive time-sensitive photos and videos known as “snaps”, which are hidden from the recipients once the time limit expires (images and videos still remain on the Snapchat server). Users can add text and drawings to their snaps and control the list of recipients which they send them [45].	2011
TikTok	Highlights bitesized looping videos that can also have musical overlays [45].	2016
TrevorSpace	An online community for LGBTQ youth ages 13 through 24 to receive advice and support from peers [46].	2008
Tumblr	Microblogging platform that allows users to post text, images, video, audio, links, and quotes to their blog. Users can also follow other blogs and repost other users’ content to their own blog [45].	2007
Twitter	Real-time social network that allows users to share 140-character updates with their following. Users can favorite and retweet the posts of other users, as well as engage in conversations using @ mentions, replies, and hashtags for categorizing their content [45].	2006
YouTube	Video sharing website to watch online videos. Users can create and upload their own videos to share with others [47].	2005
Yubo	Social media app allowing users create a profile, share their location, and flip through images of other users in their area. Users can either scroll through the current livestreams or browse individual profiles by swiping right on profiles they like and left on profiles they don’t [48].	2015
Vent	Semi-anonymous social networking app where users share their feelings without the fear of a negative backlash. Users voice their opinion to a supportive community [49].	2013

## Data Availability

Not applicable.

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
