# Peer review of "Support over Social Media among Socially Isolated Sexual and Gender Minority Youth in Rural U.S. during the COVID-19 Pandemic: Opportunities for Intervention Research"

_ijerph, 2022, doi:10.3390/ijerph192315611_

Round 1

Reviewer 1 Report

Review 

Support over social media among socially isolated sexual and gender minority youth in rural U.S.during the COVID-19 pandemic: Opportunities for intervention research

The article  deals with an important topic and questions, it is founded in relevant data and approached with an adequate method, and everything is presented with a reader friendly clarity within a clear structure, hence also providing clear results. Nonetheless, there are some weaknesses that I find it important that the authors address before the article is published. 

1.     The main weakness of the article of the is the lack of a theoretical framework that would raise the level beyond that of a mere report and clearly account for this as an academic/ scientific contribution. Why and how is this theoretically relevant? Are there I implications in relation to earlier discussions about online vs off line behavior and the possibilities to establish spaces for social support.

2.     In this form, the article risks (probably without reason) some specific forms of methodological nationalism in how it makes invisible the context of the study and referred observations. The authors need to be more clear on the limitations of their references and statements, especially when providing background to their study. Being youth, living as GSM, rural-urban distinctions, use of social media etc varies across countries and contexts and cannot be described or referred without accounting for this. 

a.     HIPAA is not necessarily a valid / relevant abbreviation for an international audience. What does it stand for? 

b.     The use of Social media platforms varies between cohorts and contexts. A short list of the ones mentioned in the article is needed with a short description of their user logic. 

3.     I did not receive the list/ table of participants’ demographics and that would have been relevant. I further wonder what the authors did in order to ensure diversity in the sample. Being a qualitative study and engaging with vulnerable groups marked by intersectionalities I assume that this is central to the quality of the study.

4.     It needs to be clarified what is meant with an reflexive thematic analysis with an experiential and realist framework since it encompasses methodological and epistemological assumptions, distinct from the method description that follows. 

5.     The results are relevant but this part should be extended in order not to risk make invisible the participants lives e.g. in terms of their need for social support, their life situation etc. This is needed also in order to confirm assumptions that the study relies on explicitly, and the higher level of vulnerabilities that GSM people are subject to and the potential relevance of social media. 

6.     Regarding limitations of the study the authors argue on the first line as if this was a quantitative study - which is not the case. The potential problems need to be addressed with regards not to the size of the sample and potential conclusions that can be made but with regards to the variations and diversity the might have missed for different reasons. Again – we deal with a case where intersectionality is highly relevant.

Reviewer 2 Report

The paper aims to analyze base on a qualitative study how the SGM youth, living in rural areas used existing social media platforms for seeking support during the first COVID-19 lockdown as well as which platform characteristics and features contributed the most to accomplish this purpose. The study aims also to outline how digital interventions could be delivered on specific social media platforms to provide support to SGM youth living in rural areas.

The design of the study, the methodology and the procedures of the data analysis, i.e. the process of formulation and interpretation of themes are clearly described. It is very informative to the readers that the interview guideline and the main themes that emerged from the interview are also included as supplementary materials. My questions/ suggestions to the authors are the following. It would be very informative to the reader to explain in more detail what is specific in the support seeking behaviors of SGM youth from the rural areas compared to their peers living in other types of residential places? Are there any specific problems that this group experiences and how they guide young people’s online social interactions and communication for seeking support during the first lockdown? I would be also interesting to describe if they prefer to seek support about problems related with their gender identity or with other problems and how the situation of lockdown impacts on this? It is interesting to comments also if the study includes an analysis of shared online contents or it focuses only on the narratives of the young social media users about their support seeking activities the social platforms?  

Overall, this is very well written paper and these questions/suggestions aim to give the reader better orientation in the social problems that SGM youth living in rural areas have experienced in the times of the first lockdown and how they were trying to come with these problems by the means of online social media.

Reviewer 3 Report

Thank you for the opportunity to review the article “Support over social media among socially isolated sexual and gender minority youth in rural U.S. during the COVID-19 pandemic: Opportunities for intervention research.” The article's subject is interesting and not so analyzed, so a study along these lines and for this demographic category is welcomed.

Overall, the subject is corrected attributed to the Global Health section of the International Journal of Environmental Research and Public Health journal. 

Foremost, it is important to emphasize that the article is well-structured, the arguments are very clearly presented, and they are logically connected with the aim posed in the first part of the article. 

Moreover, the research design is consistent with the objective of the study and the data used. Also, the research results are clear and in line with the study's objective. 

It is to be appreciated that the respondent recruitment and data collection process was passed and approved by the Institutional Review Board and is described in the article.

However, I would have liked the framework for analysis to be more explicit.

I also appreciate qualitative research that aims to address in depth certain social issues, but I think that a section of the literature review that includes related studies along these lines should be added to improve the quality of the article.

I also appreciate the perspectives developed in the Discussion section and the fact that the authors have documented the limitations of their research very well.

Just one more small comment about the abstract. I recommend reformulating it so that it coherently expresses the article's purpose, the research methods used, and the results reached (to have flow) ... In this form, the abstract seems fragmented, being structured by chapters.

Also, on line 35, spaces should be added between keywords, and those highlighted sections removed.

I congratulate the authors and consider that the study, with some minor revisions, can be published in the journal.

Round 2

Reviewer 1 Report

In my opinion the authors have sufficiently and very well clarified aspects that I raised in my first review. I find no reasons to suggest or require additional revisions.